# Effect of Vitamin D Supplementation on the Fetal Growth Rate in Pregnancy Complicated by Fetal Growth Restriction

**DOI:** 10.3390/children9040549

**Published:** 2022-04-12

**Authors:** Karolina Jakubiec-Wisniewska, Hubert Huras, Magdalena Kolak

**Affiliations:** Department of Obstetrics and Perinatology, Jagiellonian University Medical College, 23 Kopernika Street, 31-501 Krakow, Poland; karolina.jakubiec@uj.edu.pl (K.J.-W.); hubert.huras@uj.edu.pl (H.H.)

**Keywords:** fetal growth restriction, FGR, vitamin D, fetal growth rate

## Abstract

Background: Fetal growth restriction (FGR) increases the risk of intrauterine fetal death, infant death and complications in childhood, and diseases that appear in adulthood. Vitamin D may affect fetal vascular flow. The aim of the study was to check if the rate of fetal growth in pregnant women with FGR differs depending on whether the patient was supplemented with vitamin D in the recommended dose of 2000 IU, not supplemented at all, or supplemented with vitamin D in low doses. Methods: Patients were divided into two groups: suboptimal vitamin D dosage and an accurate dosage of 2000 IU. Fetal growth progress was observed for 14 days. Results: Fetal weight was higher at the beginning, after 1 and 2 weeks of observation in the optimal vit. D group compared with the suboptimal group. The analysis was adjusted to the mother’s age, gestational week, and the number of pregnancies. Conclusions: Greater fetal weight gain can be observed in women with FGR (fetal growth restriction) who intake vitamin D at the recommended dose of 2000 IU compared with women with FGR and with a vitamin D intake dosage lower than 500 IU.

## 1. Introduction

A low birth weight affects 3–10% of live-born newborns in developed countries, while it is even more prevalent in developing countries, where it affects up to 20% of the babies [1]. The single largest risk factor for stillbirth is unrecognized fetal growth restriction [2].

Fetal growth restriction is associated with intrauterine deaths, neonatal deaths (it is responsible for ca. 10% cases of perinatal deaths [3]), and complications at later stages of child’s life or diseases occurring in adulthood [4]. For a birth weight below 10 percentile for a given gestational age, the risk of neonatal death is 1.5%, and it is twice as high as in newborns of normal body weight. With the child’s weight below the fifth percentile, this risk rises to 2.5% [5].

The fetal growth can be inhibited at any period during a pregnancy. The phenomena restricting fetal growth at early stages of a pregnancy, during the first growth stage, result in an overall growth restriction, while at later stages, the development of only specific tissues, e.g., fatty or muscular tissues, is inhibited, while other organs, such as the brain and the heart, are spared [6]. In 2004, Thorton proposed a classification distinguishing between early- (type I) and late-onset fetal growth restriction, which differ in their etiology, course and management. The 32–34 gestational week was assumed as a boundary for these two FGR types [7,8,9]. 

Epidemiological studies suggest that the prevalence of vitamin D3 deficiency reaches 90% in the world. Detailed data concerning the Polish population is not available. An attempt was made to estimate vitamin D levels in a large study in 5775 volunteers living in rural areas. Reduced vitamin D levels, below 30 ng/mL, was found in nearly 90% subjects [10]. These findings are consistent with data for the global population. 

In pregnancy, vitamin D deficiency is common all over the world, even with routine use of prenatal vitamins [11], because a dose of this vitamin in popular multivitamin formulations is too low to sufficiently increase its serum levels [12]. According to guidelines for the supplementation of vitamin D in Central Europe from 2013, the supplementation of vitamin D at a dose of 1500–2000 IU/day (37.5–50 g/day) is recommended for all pregnant women, and it should start at least in the second trimester of pregnancy; additionally, the gynecologist/obstetrician should recommend starting supplementation soon after the pregnancy is confirmed. Furthermore, a periodic monitoring of serum 25(OH)D levels is recommended, if possible, to establish an optimal dose and to verify the effectiveness of the supplementation. Its aim is to achieve and maintain 25(OH)D at a level of 30–50 ng/mL (75–125 nmol/L). The guidelines precisely establish diagnostic criteria characterizing the body vitamin D supply: vitamin D deficiency was specified as the 25(OH)D level <20 ng/mL (<50 nmol/L)], the level of 20–30 ng/mL (50–75 nmol/L) was considered suboptimal 25(OH)D supply, and the level of 30–50 ng/mL (75–125 nmol/L) was considered a target for ensuring the pleiotropic effect of vitamin D [13]. According to the Recommendation of the Polish Society of Gynecologists and Obstetricians for vitamin supplementation, a daily vitamin D dose for pregnant women should amount to 2000 IU.

Recently, the number of reports on the role that vitamin D plays during pregnancy, including a pregnancy complicated by intrauterine growth restriction, has been increasing. Vitamin D probably plays an important role in fetal growth. It was found that the human placenta contains elements involved in vitamin D signaling pathways, i.e., the vitamin D receptor (VDR) and the CYP27B1 enzyme (1 alpha-hydroxylase) activating this vitamin [14]. The active form of vitamin D, acting through VDR and the cAMP/protein kinase signaling pathway, regulates the expression and secretion of human chorionic gonadotropin in the syncytiotrophoblast [15] and increases the production of steroids in the placenta [16]. Vitamin D is also important in glucose and insulin metabolism [17], so it can play a role in ensuring glucose availability for transport through the placenta and its subsequent use by a fetus. As a regulator of calcium homoeostasis and transport, calcitriol can also directly influence fetal growth, through its influence on development of skeletal muscles and bones [18].

A relationship between vitamin D supplementation in pregnancy complicated by FGR and the growth still remains controversial. Although numerous reports suggest a positive effect of supplementation with this vitamin, some of them do not confirm this relationship or give inconclusive results [19]. Taking into account the significance of the problem of intrauterine growth restriction and the fact that practically no proven methods for the treatment of this dysfunction are available, a search for substances that may effectively support the management of FGR is very important or even necessary, especially for substances that, when used, can be practically only beneficial, i.e., an improvement in health and reductions in fetal and neonatal mortality, without a risk of adverse reactions. Hence, we attempt to determine the effect of vitamin D supplementation on the growth rate of a fetus with FGR.

The aim of this study was to verify whether the fetal growth rate in pregnant women diagnosed with early fetal growth restriction differs depending on whether the patient supplements vitamin D at the dose recommended by the Polish Society of Gynecologists and Obstetricians, 2000 IU; does not supplement vitamin D at all; or takes vitamin D at a low dose, i.e., below 500 IU.

## 2. Materials and Methods

Prospective cohort study: a group of patients with early FGR was divided into two subgroups according to the amount of vitamin D supplemented. The Ethics Committee approved this study (No. 122.6120.262.2016 on 29 September 2016).

Inclusion criteria included the following:Presence of type 1 fetal growth restriction (early FGR);Single pregnancy;Before the end of the 32nd gestational week;Age 18–45 years old; andExpected to give birth between October and March, inclusive, as during this period no skin synthesis of vitamin D occurs, which could affect the measurement results.

Exclusion criteria included the following (at least one of the following must be met):Lack of patient consent to participate in the study;Diabetes, hypertension, or hypothyroidism diagnosed at any time (because these diseases are risk factors for FGR);Vegetarian/vegan diet;Smoking; andThe presence of a clear cause known to inhibit the fetal growth potential, e.g., fetal genetic disorders diagnosed earlier.

The study was conducted at the Obstetrics and Perinatology Clinical Department, Collegium Medicum, Jagiellonian University in Kraków, from October 2017 to March 2019. The clinical part, i.e., ultrasound measurements, were performed at the Ultrasound Department, the Pregnancy Pathology Unit of the Obstetrics and Perinatology Clinical Department. The laboratory tests were performed at two locations. The test samples for determining vitamin D level were collected during routine monitoring examinations at the Pregnancy Pathology Unit or the Pregnancy Pathology Outpatients Clinic. Then, the appropriately fixed material was transported to the Clinical Biochemistry Department, Collegium Medicum, Jagiellonian University in Kraków, where the vitamin D level was determined using the ELISA method.

The studied population covered women meeting the inclusion criteria, i.e., women with a single pregnancy, before the end of the 32nd gestational week at the moment of enrolment into the study, aged 18–45 years, and diagnosed with early-onset fetal growth restriction according to the standard on a basis of the Delphi consensus criteria. Patients with an estimated fetal weight below the 10th percentile and UA PI below the 95th percentile; with extremely low estimated fetal weight, i.e., below 3 percentile; or with AEDF (absent end of diastolic flow) found in the umbilical artery were included in the study. The patients were referred to the Pregnancy Pathology Outpatients Clinic or the Pregnancy Pathology Unit, Collegium Medicum, Jagiellonian University, by a doctor in charge of their case due to suspected FGR, and there, they received an outpatient consultation or were hospitalized. Women coming to the clinic between October and March, inclusive, were enrolled in the study, as during this period, no skin synthesis of vitamin D occurs, which could affect the measurement results. Both during the study, lasting 14 days, and after its completion, they were constantly monitored at the Pregnancy Pathology Outpatients Clinic or the Pregnancy Pathology Unit, CMUJ.

Participation in the study depended on patient consent; before the interview, the collection of material for the laboratory tests, and the ultrasound scan, the patient received information about participation in the study, specifically detailed information on the aims and rules for conducting the study, the expected benefits, and the risks associated with participation in the study, and signed an informed consent form to participate in the study, containing statements acknowledging that participation in the study was voluntary after reading the information provided about participation in the study, that they could ask the investigator questions and receive answers to those questions, and that they could withdraw from the study at any stage. Each patient was asked to complete a questionnaire concerning what types of vitamin supplement they took and how often they take it; the questionnaire is attached in the Appendix A.

One hundred patients were enrolled in the study. The subjects were divided into subgroups on the basis of interviews concerning vitamin D supplementation by the patient, i.e., information from the completed questionnaire. The patient did not have to know what dose of vitamin D they took; they only needed to provide the name of the vitamin formulation taken at that time, and the frequency and regularity of taking the supplement. The vitamin D dose in a given formulation was established from the online Medicine Register. The first study subgroup (*n* = 50) consisted of patients who, at enrolment, took vitamin D at a dose of 2000 IU specified in the Recommendation of the Polish Society of Gynecologists and Obstetricians for supplementation of vitamins. The second study group (*n* = 50) consisted of patients who did not take vitamin D at all or supplemented vitamin D in a low dose, i.e., below 500 IU.

### 2.1. Study Methods, Procedures, Measurements, and Tools

Each patient was monitored for 14 days. The clinical part of the study included an ultrasound scan. The scan is a basic method used to confirm correct fetal development or to diagnose its disruptions. The scans were performed twice two weeks apart and, in some patients, three times one week apart. The scan involved standard measurements of fetal biometric parameters, including biparietal diameter (BPD), head circumference (HC), abdominal circumference (AC), and femur diaphysis length (FL), and the ultrasound unit used those measurements to calculate an estimated fetal weight (EFW). During the measurements, Hadlock 2 formulae were used as it is characterized by the greatest accuracy. The gestational age of the patient at the moment of the ultrasound scan was not of great importance, as between the 24th and 35th gestational weeks, daily fetal growth occurs at a similar level. Furthermore, the estimated weight was transformed into percentiles by the ultrasound unit, and this was taken into account in statistical calculations to eliminate the importance of the gestational age. The ultrasound measurements were performed at the Prenatal Ultrasound Diagnostics Laboratory, the Pregnancy Pathology Unit, Obstetrics and Perinatology Clinical Department, using the same model as the ultrasound unit: Samsung HS60.

All patients covered by the study had one blood sample collected to evaluate initial vitamin D levels. The blood was collected during the collection of blood for routine monitoring tests performed for each patient coming to the Pregnancy Pathology Outpatients Clinic or admitted to the Pregnancy Pathology Unit. Then, appropriately fixed material was transported to the Clinical Biochemistry Department, CM UJ, where the vitamin D level was determined using the enzyme-linked immunosorbent assay (ELISA). An inactive metabolite of vitamin D3, 25(OH)D was determined. It is a standard procedure for determining vitamin D3 resources in a body.

Both studied subgroups received the general standard management. They did not receive LMWH, L-arginine, or other therapy-supporting experimental methods. No clinical intervention was used during the study.

### 2.2. Statistical Analysis

Data were collected, analyzed, and elaborated using the STATISTICA 12 software (StatSoft, Tulusa, Oklahoma). The results were listed in tables and illustrated in diagrams. The nominal variables were summed as sizes and percentage rates. Depending on the distribution of variables, continuous features were presented as means with standard deviation (SD) or as medians with interquartile range (IQR). The nominal variables were compared using Pearson’s chi-squared test or Fisher’s exact test when over 20% of cells had expected sizes below 5. In the case of multivariate variables measured on the ordinal scale (e.g., pregnancy number), the Cochran–Armitage test for trend was applied. When a given variable assumed many values, the Mann–Whitney U test was used. The distribution normality in the groups was evaluated with the Shapiro–Wilk test, while homoscedasticity was assessed with Levene’s test. Differences between the continuous variables of normal distribution found during comparisons of the two groups were tested with Student’s t-test or Welch’s t-test, depending on an assumed homoscedasticity. For variables with distribution significantly deviating from the normal, the Mann–Whitney U test was used. In this study, the Pearson correlation coefficient (for variables with normal distribution) and the Spearman’s rank correlation coefficient, describing a monotonical relation, were used to evaluate the relationship between the continuous variables. Linear regression methods were used to assess the multivariate influence of variables on selected continuous features. The standard assumed significance level for all analyses is 0.05. All hypotheses are two-sided hypotheses.

## 3. Results

In the study group, the data of the patients supplementing vitamin D at the dose of 2000 IU recommended by the Polish Society of Gynecologists and Obstetricians (*n* = 50) were compared with the data of the patients supplementing vitamin D at a dose found in the majority of supplements for pregnant women, i.e., below 500 IU (*n* = 50). These groups were diversified (*p* = 0.0002 in terms of the gestational week. In the group of patients supplementing vitamin D at the dose of 2000 IU, the gestational week median was 31.57 (IQR: 29.82;32.86), while in the group supplementing vitamin D at the dose below 500 IU, it was 29.57 (IQR: 25.43;31.46). No significant differences were observed for age, the number of previous pregnancies, or the regularity of supplementation. The vitamin D level was significantly higher (*p* < 0.0001) in the group taking the dose of 2000 IU versus the group of patients supplementing vitamin D at the dose below 500 IU, with medians of 36.18 (IQR: 26.48;45.26) and 23.04 (IQR: 16.03;29.25), respectively. The vitamin D deficiency was more common in the group of patients supplementing vitamin D at the dose below 500 IU (40.00% vs. 8.00%, *p* = 0.0002). The results are shown in Table 1.

### Fetal Weight Depending on a Dose of Supplemented Vitamin D

Due to the initial imbalance in the patient group in terms of baseline characteristics, where a significant difference was observed for the gestational week (*p* = 0.0002), or to the potential impact of mother’s age on the analyzed parameters (*p* = 0.0580), we decided to present the results of the multivariate analyses. Mother’s age, gestational week, number of pregnancies, and vitamin D dose were included in models predicting fetal weight parameters. The initial fetal weight mainly depended on the gestational week (*p* < 0.0001) but did not depend significantly on the vitamin D dose (*p* = 0.3956). The significance of the number of previous pregnancies was close to statistically significant (*p* = 0.0773).

The average fetal weight measured on Day 7 of the observations in women taking vitamin D at the dose of 2000 IU was higher by 91.839 g (95% CI: 10.30; 173.38) than in the group of women taking vitamin D at the dose below 500 IU (*p* = 0.0277), adjusted to confounding factors (mother’s age, gestational week, and number of pregnancies).

The average fetal weight measured on Day 14 of the observations in women taking vitamin D at the dose of 2000 IU was higher by 156.051 g (95% CI: 64.37; 247.74) than in the group of women taking vitamin D at the dose below 500 IU (*p* = 0.0011) for assumed values of the remaining factors (mother’s age, gestational week, and the number of pregnancies). The results are shown in Table 2.

The fetal weight increase in the patients taking vitamin D at the dose of 2000 IU was also significantly higher than in the patients taking vitamin D at the dose of <500 IU. The increase in fetal weight was significantly higher both after one week and two weeks of observations and was higher, on average—59.43 g (95% CI: 35.79;83.06, *p* < 0.0001) and 141.01 g (95% CI: 98.41; 183.61, *p* < 0.0001), respectively—for assumed values of the remaining factors (mother’s age, gestational week, and the number of pregnancies). The results are shown in Figure 1.

With an increase in the vitamin D level by 1 ng/mL within 14 days, the fetal weight increased by 2.152 g on average (95% CI: 0.2685;4.0354), with the values of other parameters (mother’s age, gestational week, and the number of (previous) pregnancies) specified (Table 3).

A positive relationship was observed between vitamin D and a difference in weight (between Day 7 and Day 14 and the baseline value) of Spearman’s rank correlation coefficient (r_s_) = 0.3431 (*p* = 0.0005) and r_s_ = 0.3114 (*p* = 0.0025), respectively, as well as a difference in the weight percentile (between Day 7 and Day 14 and the baseline value) of rs = 0.3764 (*p* = 0.0001) and r_s_ = 0.3662 (*p* = 0.0003), respectively. The results are shown in Table 4 and in Figure 2 and Figure 3.

## 4. Discussion

Fetal growth restriction is a serious problem in modern obstetrics and a subject of intense clinical studies due to the frequency of associated complications and a lack of an effective treatment method. Newborns affected by this pathology are characterized by higher perinatal mortality, as well as mortality and morbidity later in life. 

Already during labor, these newborns are at a higher risk of meconium aspiration syndrome, and more frequently present with hypoxia and acidosis symptoms in the perinatal period [20]. The most common complication is respiratory distress, as well as bronchopulmonary dysplasia and respiratory infections, resulting not only from lung immaturity but also from long-term use of a ventilator. Furthermore, they more frequently develop necrotizing enterocolitis (NEC) and hematological disorders, such as polycythemia, leukopenia, or thrombocytopaenia [21,22,23,24]. Perinatal asphyxia and premature birth may be causes of neurological complications, such as cerebral palsy or cognitive disorders. Perinatal asphyxia causes neurological disorders in fetuses with FGR more frequently than in fetuses with normal body weight [25,26]. There is an opinion that even newborns with FGR born at term are characterized by lower intelligence, have worse school results, and have problems with their behavior [27,28,29,30]. FGR often correlates with hypertension, preeclapsia, eclampsia and HELLP syndrome, those complications increase maternal mortality and morbidity [31]. 

Considering the importance of the problem of fetal growth restriction and the fact that there are practically no proven treatment methods for this disorder, the search for a substance that can effectively support the FGR therapy is very important or even necessary, especially for substances that, when used can be only beneficial, i.e., an improvement in health and a reduction in fetal and neonatal mortality, without a risk of adverse reactions. An attempt to determine the effect of vitamin D supplementation on a growth rate of a fetus with FGR and on the improvement of cerebral and umbilical flows (which is equal to an improvement in fetal well-being) is fully justified, especially since indications of its positive effects have been appearing in scientific reports for several years.

A relationship between vitamin D supplementation in pregnancy complicated by FGR and fetal growth still remains controversial. Although numerous reports suggest a positive effect of supplementation with this vitamin, some either do not confirm this relationship or give inconclusive results.

The research shows that low levels of vitamin D may contribute to a lack of normal fetal growth, and in some studies, it was found that the risk of fetal hypotrophy in mothers with low levels of this vitamin is several times higher [19,32,33]. 

One of the first studies on vitamin D supplementation during pregnancy was the study performed by Brooke et al. in 1980, which found that SGA was more frequent in newborns of mothers receiving placebos than in newborns of mothers receiving 1000 IU of vitamin D in the third trimester of pregnancy [34]. Mannion et al. studied the relationship between the amount of vitamin D taken by pregnant women and the child’s growth. They demonstrated that each successive 40 IU of vitamin D taken by the mother was associated with an increase in the birth weight by 11 g [34]. In the study involving 4000 pregnant women, it was demonstrated that newborns of mothers with lower vitamin D levels were characterized by lower body weight and length, and a higher risk of SGA [33]. It is suggested that vitamin D supplementation influences reduced incidence of FGR [35]. 

However, not all studies confirm a relationship between vitamin D and the growth of a newborn. In their study, Bodnar et al. demonstrated that there was no relationship between mother’s vitamin D levels and a risk of SGA development in black women, and a lower risk of SGA in white women with suboptimal vitamin D levels [19].

As vitamin deficiencies during pregnancy are common all over the world, it is important to raise awareness in patients who are pregnant and taking vitamin supplements. Frequently, patient because of lack of knowledge, possibly due to financial reasons, advertisements, or suggestions from a pharmacy, they purchase vitamin formulations with incomplete compositions or inconsistent with the recommendations.

In the study group, patients supplementing vitamin D in a recommended dose of 2000 IU were slightly older, with a median age of 29 years, while patients taking vitamin D in low doses were younger, with a median age of 28 years. The result was close to being statistically significant (*p* = 0.0580). It can be assumed that older patients paid more attention to a need to take this vitamin at a dose recommended by the Polish Society of Gynecologists and Obstetricians. Older women significantly more often took vitamin D regularly and were enrolled into the study at the later period of their pregnancy, r = 0.2878 (*p* = 0.0037). 

Patients supplementing vitamin D at the dose of 2000 IU and below 500 IU differed in the gestational age at the moment of their inclusion in the study (31w4d vs. 29w5d). The youngest gestational ages were 25 weeks and 4 days in the group of patients supplementing vitamin D at the dose of 2000 IU and 24weeks and 0 days in the group of patients supplementing vitamin D at the dose below 500 IU. It can be assumed that, in the group supplementing vitamin D in lower doses, the problem of restricted fetal growth was noticed earlier, and this most likely is tantamount to its earlier onset. Patients of up to and including the 32nd gestational week were included in the study due to a very similar fetal growth rate up to that gestational week and because the 32nd gestational week is the upper limit for a diagnosis of early-onset FGR (it is diagnosed up to the 32nd gestational week). Early-onset FGR represents up to 30% of all cases of intrauterine growth restriction, but it is easier to diagnose because it is associated with more severe placental insufficiency and more frequent cases of abnormal vascular flow, especially in the umbilical artery [36].

No significant differences in the number of previous pregnancies were observed between the subgroups, with primigravidae significantly predominating in both of them, representing 66% of the patients. It is consistent with the position of the American College of Obstetricians and Gynecologists (ACOG) and Royal College of Obstetricans and Gynecologists RCOG, which specify nulliparity as a risk factor for FGR [37]. Patients during their second pregnancy with one childbirth in the interview represented 24% of the group, and patients with two and with three childbirths in the interview amounted to 6% and 4% of the group, respectively. 

After 7 days, eight patients withdrew from the study due to a deterioration in the intrauterine fetal condition. Intrauterine fetal death occurred in three of them, while five required emergency cesarean section due to deteriorated vascular flows and abnormal cardiotocography patterns. All of them were in the subgroup receiving vitamin D at a dose below 500 IU. This results in intrauterine mortality in the studied population at a level of 3%. Fetal growth restriction is responsible for the majority (66.4%) of intrauterine deaths [38]. All five newborns born by an emergency cesarean section survived the first day, and their further outcome has not been monitored.

To assess fetal growth in the study group, all patients (excluding those who did not complete the study) had three ultrasound scans each: at the moment of inclusion into the study, and after 7 and 14 days. In the case of FGR, a repeated ultrasound assessment is recommended after 14 days for reliable evaluation of growth dynamics, but because the majority of patients were hospitalized and had a routine ultrasound scan with biometric measurements performed every week, it was decided to include the measurements made after 7 days into the calculations. A different aspect is a need for earlier monitoring in the case of specific flow anomalies found, which when left unattended for 2 weeks, may deteriorate and result in an intrauterine death of the fetus. According to Figueras and Gratacos, monitoring should be performed once a week in the cases of CPR < 1, UA PI >95 percentile, and MCA PI < 5 percentile; twice a week in the case of AEDV in UA; and every day for REDV (CPR—cerebral placental ratio, UA—umbilical artery, MCA—middle cerebral artery, AEDV—absent end of diastolic velocity, REDV—reversed end-diastolic velocity, and PI—pulsatility index). 

To minimize the margin of error, all scans were performed on the same ultrasound unit and by the same person. For Doppler measurements of vascular flows, in nearly every case, the measuring gate was positioned perpendicularly and the angle of insonation was nearly zero. For measurements that are technically difficult because of the fetus position, concerning especially the middle cerebral artery, the angle of insonation did not exceed 30 degrees. 

In the subgroups of patients taking vitamin D at doses of 2000 IU and of 500 IU, fetal weights at the moment of inclusion into the study differed significantly (1360 g vs. 956 g); this, to a large extent, also resulted from differences in the pregnancy duration in those subgroups. A difference was also found when the percentiles of those weights were compared (3 vs. 1.3). However, during a multivariate analysis, taking into account the mother’s age, the gestational week, the number of previous pregnancies, and the dose of vitamin D did not differ, but higher fetal weights were observed both after one and two weeks of observations for women supplementing vitamin D at the dose of 2000 IU, by 92 g and 156 g, respectively. The same situation occurred for weight increases, which were higher in women supplementing the vitamin at recommended doses, by 59 g after one week and 141 g after two weeks; in the correlation tests, the higher the vitamin D level, the higher the weight increase, both after one and two weeks of observations. It is consistent with scientific reports in which fetal weight was positively correlated with vitamin D levels in the mother [31]. In one large study in a group of pregnant women, with early-onset pre-eclampsia, significantly lower levels of vitamin D were also found in cases additionally complicated by FGR, when compared with the subgroup with normal neonatal weight. The birth weight percentile was higher for high vitamin D levels [39]; however, in pregnancies not complicated by hypertension or pre-eclampsia, a relationship between low vitamin D levels and low birth weight of newborns was observed. In a study covering 1198 American women, it was found that the birth weight of a newborn was lower in the case of low vitamin D levels in the patient, and the risk of the low birth weight decreased with the increase in the mother’s serum vitamin D levels to the upper limit of the normal range [19]. In a study covering over 7000 women, it was demonstrated that the restricted fetal growth in the third trimester more frequently occurred in patients with vitamin D deficiency; furthermore, after birth, the children were characterized by lower weight, smaller head circumference, and body length. In the case of 25(OH)D deficit below 20 ng/ml, the risk of premature birth and FGR increased significantly and reached 17.3% and 22.6%, respectively. 

The effect of vitamin D on fetal growth, postulated in studies, is multidirectional, i.e., affecting directly fetal bones and muscles [40,41], and indirect, e.g., through the role in glucose and insulin metabolism [42,43]. With these very extensive functions occurring within the body of the child developing in the uterus and in fetuses from physiological pregnancies, the lower fetal weight resulting from its deficiency should rather be an expected result.

In the multivariate analysis of the study group, it was found that the fetal body weight was lower in primigravidae. It is also consistent with the fact that nulliparity represents a risk factor for low fetal and birth weight. Numerous reports indicate that children of primigravidae are characterized by higher birth weight. According to Toulon and Combet et al., the average birth weight at the time of birth in primigravidae is higher by 3.4% than in primigravidae [44].

In the study group of the patients supplementing vitamin D at the dose of 2000 IU, the vitamin D level was higher compared with those supplementing vitamin D at the dose below 500 IU (36.18 ng/mL vs. 23.04 ng/mL). In the group taking low doses, the vitamin levels were abnormal in 76% of women in the study group. It is therefore consistent with the logic that, when the doses of the supplement taken are several times higher, we expect higher serum levels, and it is also consistent with scientific reports. In the study by Hollis et al., significant differences were found between achieved serum 25(OH)D levels in the groups receiving 400 IU vs. 2000 IU and 400 IU vs. 4000 IU, while significant differences in the groups taking 2000 IU vs. 4000 IU occurred only in the 24th and 32nd gestational weeks [36]. Assessing the effectiveness of vitamin D supplementation at doses of 400 IU vs. 2000 IU, Zerofsky et al. demonstrated that the achieved serum 25(OH)D level was significantly higher and achieved planned values only in the group receiving 2000 IU [45].

When the mother’s age, regularity of supplementation, and vitamin D dose were taken into account, its levels in the study group were lower by 2.71 units, on average, than in patients from the control group. It can be concluded that, regardless of the vitamin D dose taken, it is lower by three units on average in the group of patients with intrauterine fetal growth restriction. The patients with pregnancy complicated by FGR had higher vitamin D deficiencies than women with physiological pregnancies, despite vitamin D supplementation in the same dose and with the same regularity. We can attempt to conclude that the low vitamin D level belongs to risk factors for abnormal fetal growth, and in patients from the risk group or with suspected FGR, supplementation at doses even higher than the recommended 2000 IU should be initiated to increase the serum vitamin D levels, which are lower despite taking the same dose of the supplement. Dawodu et al. demonstrated that the administration of vitamin D in the doses of 2000–4000 IU during pregnancy in women with a high endemic risk is safe and that the dose of 4000 IU is the most effective one in optimizing 25(OH)D levels in a mother and a child [46].

### Possible Limitations of the Study

The greatest limitation of this study, which may affect the results, is the interview conducted with the patient, which consisted of a survey about taking vitamin D as well as an oral interview about factors that may potentially influence FGR development in pregnant women. In the questionnaire, the patient provided the name of the formulation they took and how often they took it, but verification of this information was impossible. Another issue remains in the oral interview on the patient’s medical history or smoking. A significant percentage (even up to 50%) of smoking women do not admit to smoking because they are afraid of being criticized.

The next limitation of this study is a lack of follow up after birth. Part of the patients were from an outpatient clinic and delivered in another hospital. There were no data about delivery, birth weight, and newborn conditions collected.

## 5. Conclusions

A higher increase in fetal body weight is observed in women with pregnancies complicated by FGR who take vitamin D supplementation at the dose of 2000 IU than in the group of women with pregnancies complicated by FGR who take vitamin D supplementation at a dose below 500 IU.

## Figures and Tables

**Figure 1 children-09-00549-f001:**
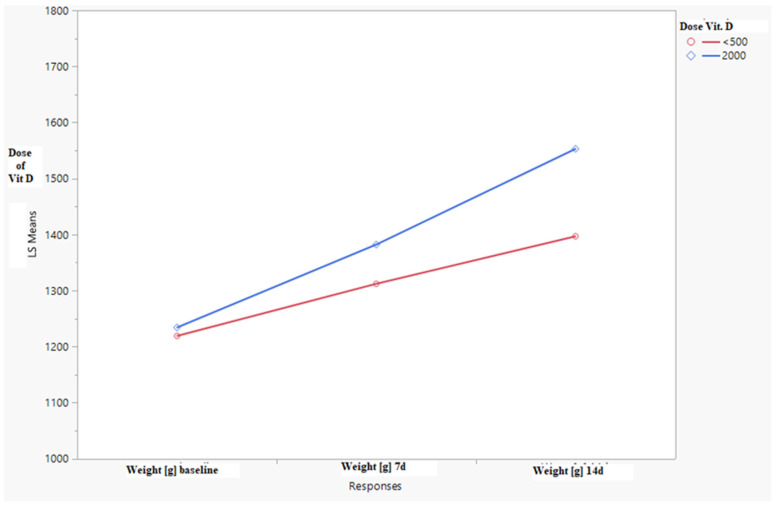
Fetal weights at the moment of inclusion into the study, after 7 and 14 d of observation depending on vitamin D dose.

**Figure 2 children-09-00549-f002:**
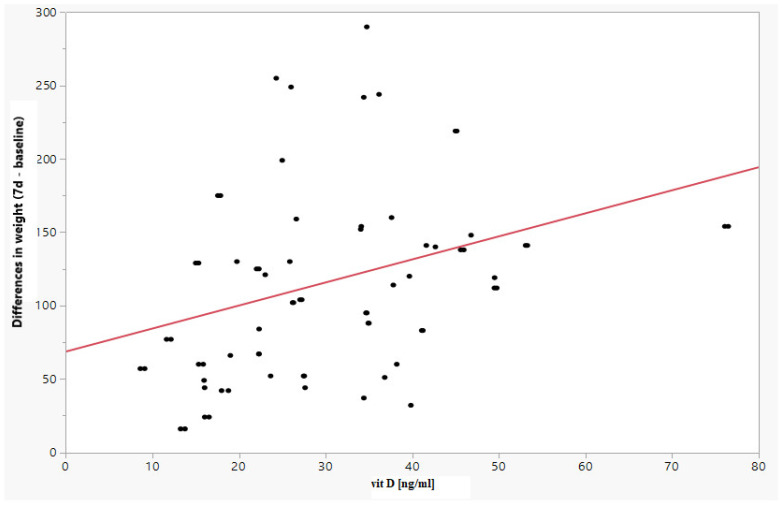
Increase in fetal weight on 7days of observations depending on the vitamin D level.

**Figure 3 children-09-00549-f003:**
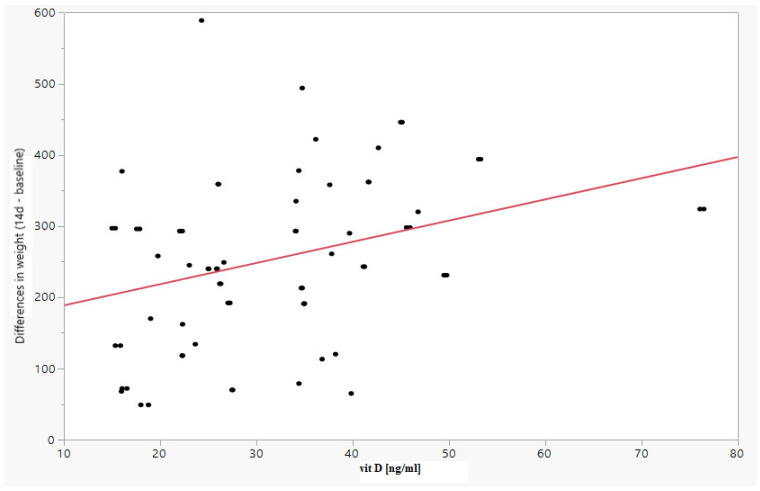
Increase in fetal weight on Day 14 of observations depending on the vitamin D level.

**Table 1 children-09-00549-t001:** Comparison of baseline characteristics of patients supplementing vitamin D at the dose of 2000 IU recommended by the Polish Society of Gynecologists and Obstetricians versus patients supplementing vitamin D at a dose found in the majority of supplements for pregnant women, i.e., below 500 IU.

Variable	Measure/Category	<500 (*n* = 50)	2000 (*n* = 50)	Total (*n* = 100)	*p*
Mother age (years)	Me(Q1;Q3)	28.00 (25.50;32.00)	29.00 (27.75;35.25)	29.00 (27.00;34.00)	0.0580
Gestational week	Me(Q1;Q3)	29.57 (25.43;31.46)	31.57 (29.82;32.86)	31.07 (27.57;32.14)	**0.0002**
Pregnancy	1	32 (64.00%)	34 (68.00%)	66 (66.00%)	0.3055
	2	12 (24.00%)	12 (24.00%)	24 (24.00%)	
	3	2 (4.00%)	4 (8.00%)	6 (6.00%)	
	4	4 (8.00%)	0 (0.00%)	4 (4.00%)	
Vit. D (ng/mL)	Me(Q1;Q3)	23.04 (16.03;29.25)	36.18 (26.48;45.26)	27.59 (22.10;38.22)	**<0.0001**
vit. D level	Deficiency (<20 ng/mL)	20 (40.00%)	4 (8.00%)	24 (24.00%)	**<0.0001**
	Low (20–30 ng/mL)	18 (36.00%)	10 (20.00%)	28 (28.00%)	
	Normal (30–100 ng/mL)	12 (24.00%)	36 (72.00%)	48 (48.00%)	
vit.D < 30 ng/mL	<30	38 (76.00%)	14 (28.00%)	52 (52.00%)	**<0.0001**
	≥30	12 (24.00%)	36 (72.00%)	48 (48.00%)	
vit.D < 20 ng/mL	<20	20 (40.00%)	4 (8.00%)	24 (24.00%)	**0.0002**
	≥20	30 (60.00%)	46 (92.00%)	76 (76.00%)	
vit. D regularly	Yes	44 (88.00%)	48 (96.00%)	92 (92.00%)	0.2687
	No	6 (12.00%)	2 (4.00%)	8 (8.00%)	
vit. D per week	Me(Q1;Q3)	7.00 (7.00;7.00)	7.00 (7.00;7.00)	7.00 (7.00;7.00)	0.1683

Notes: Q1—quartile 1, Q3—quartile 3; Bolded results are statistically significant.

**Table 2 children-09-00549-t002:** Summary of the results of the linear regression explaining selected fetal weight parameters.

Model/Dependent Variable	Parameter	(95% CI)	*p*-Value	Statistics	Value
Model: Baseline weight (g)			<0.0001	*R^2^*	87.67%
Intercept	−2752.1427	(−3120.7713; −2383.5140)	**<0.0001**	Adjusted R^2^	87.15%
Mother age (years)	1.0712	(−7.1529; 9.2952)	0.7965	*n*	100
Gestational week	131.8602	(120.4860; 143.2345)	**<0.0001**		
Pregnancy	−45.6076	(−96.3064; 5.0913)	0.0773		
Vit. D dose (<500)	−16.2069	(−53.9107; 21.4969)	0.3956		
Model: Weight 7d (g)			**<0.0001**	R^2^	87.46%
Intercept	−2796.3670	(−3194.9684; −2397.7655)	**<0.0001**	Adjusted *R^2^*	86.93%
Mother age (years)	1.8306	(−7.0622; 10.7233)	0.6837	*n*	100
Gestational week	136.9261	(124.6271; 149.2252)	**<0.0001**		
Pregnancy	−55.2567	(−110.0778; −0.4356)	**0.0482**		
Vit. D dose (<500)	−45.9195	(−86.6890; −5.1500)	**0.0277**		
Model: Weight 14d (g)			**<0.0001**	*R^2^*	86.71%
Intercept	−2753.1960	(−3196.6914; −2309.7007)	**<0.0001**	Adjusted *R^2^*	86.10%
Mother age (years)	0.5938	(−9.9168; 11.1045)	0.9108	*n*	92
Gestational week	141.0247	(127.1005; 154.9489)	**<0.0001**		
Pregnancy	−49.4031	(−110.9624; 12.1562)	0.1143		
Vit. D dose (<500)	−78.0255	(−123.8682; −32.1828)	**0.0011**		
Model: Weight difference 7d – baseline			**<0.0001**	*R^2^*	38.98%
Intercept	−44.2243	(−159.7515; 71.3029)	0.4492	Adjusted *R^2^*	36.41%
Mother age (years)	0.7594	(−1.8180; 3.3368)	0.5600	*n*	100
Gestational week	5.0659	(1.5012; 8.6305)	**0.0058**		
Pregnancy	−9.6491	(−25.5380; 6.2397)	0.2310		
Vit. D dose (<500)	−29.7126	(−41.5288; −17.8963)	**<0.0001**		
Model: Weight difference 14d – baseline			**<0.0001**	R^2^	47.71%
Intercept	17.0129	(−189.0350; 223.0609)	0.8700	Adjusted *R^2^*	45.31%
Mother age (years)	0.2289	(−4.6543; 5.1122)	0.9260	*n*	92
Gestational week	8.0611	(1.5919; 14.5303)	**0.0152**		
Pregnancy	−13.8578	(−42.4582; 14.7427)	0.3382		
Vit. D dose (<500)	−70.5059	(−91.8044; −49.2074)	<0.0001		

Notes: CI—confidence interval; Bolded results are statistically significant.

**Table 3 children-09-00549-t003:** Summary of the influence of vitamin D levels and doses on fetal weight.

Model#/Variable	Estimate	(95% CI)	*p*-Value	Statistics	Value
Model 1:			**<0.0001**	*R^2^*	26.07%
Intercept	−287.5834	(−515.0841; −60.0827)	**0.0138**	Adjusted *R^2^*	22.68%
Mother age (years)	4.1860	(−1.4094; 9.7814)	0.1406	Observations	92
Gestational week	12.9580	(5.5110; 20.4051)	**0.0008**		
Pregnancy	−31.1099	(−64.6149; 2.3952)	0.0684		
Vit. D (ng/mL)	2.1520	(0.2685; 4.0354)	**0.0256**		
Model 4:			**<0.0001**	*R^2^*	47.71%
Intercept	17.0129	(−189.0350; 223.0609)	0.8700	Adjusted *R^2^*	45.31%
Mother age (years)	0.2289	(−4.6543; 5.1122)	0.9260	Observations	92
Gestational week	8.0611	(1.5919; 14.5303)	**0.0152**		
Pregnancy	−13.8578	(−42.4582; 14.7427)	0.3382		
Vit. D dose (<500)	−70.5059	(−91.8044; −49.2074)	**<0.0001**		

Notes: R-risk. Bolded results are statistically significant.

**Table 4 children-09-00549-t004:** The Spearman’s rank correlation between the vitamin D value and other parameters.

By Variable	Spearman’s Correlation	Spearman’s Correlation *p*-Value
Weight difference, percentile (7 d− baseline)	0.3764	**0.0001**
Weight difference, percentile (14 d − baseline)	0.3662	**0.0003**
Weight percentile 7d	0.3462	**0.0004**
Weight difference (7 d − baseline)	0.3431	**0.0005**
Weight difference (14 d − baseline)	0.3114	**0.0025**
Weight percentile 14 d	0.2999	**0.0037**
Weight (g) 7 d	0.2341	**0.0191**
Weight (g) baseline	0.2012	**0.0448**
Weight percentile baseline	0.1917	0.0560
Weight (g) 14 d	0.1848	0.0778

Bolded results are statistically significant.

## Data Availability

Not applicable.

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
