# Peer review of "Effect of Vitamin D Supplementation on the Fetal Growth Rate in Pregnancy Complicated by Fetal Growth Restriction"

_children, 2022, doi:10.3390/children9040549_

Round 1

Reviewer 1 Report

Review of article “Effect of vitamin D supplementation on the fetal growth rate in pregnancy complicated with fetal growth restriction” authors Karolina Jakubiec-Wisniewska , Hubert Huras , Magdalena Kolak *
In this paper the authors explain well the usefulness of the study.  Several epidemiological studies suggest an association between vitamin D deficiency (VDD) and fetal intrauterine growth restriction (IUGR). One study explored the mechanism through which VDD induced fetal IUGR. The objective of this study is to check if the rate of fetal growth in pregnant women with FGR differs depending on whether the patient supplemented with vitamin D in the recommended dose of 2000 IU, not supplemented at all or supplemented with vitamin D in low doses. They divided patients into two groups, with suboptimal vitamin D dosage (500 IU) and with accurate dosage 2000 IU. During 14 days fetal growth progress were observed. They discovered that fetal weight was higher after 1 and 2 weeks of observation in optimal vit. D group compared with suboptimal group. This result is very interesting, as there are no similar studies on the subject, or at least there are no studies that have proved an increase in growth in the better supplemented population. The study methodology is very linear and the text is understandable and easy to read. The conclusions are perfectly in line with the objectives that the authors had set.
The authors present a new possibility to  improve IUGR growth. 
Conclusions
This is a new possibility to improve IUGR growth

Reviewer 2 Report

This is a nice study that aimed to detect whether vitamin D supplementation influences the rate of fetal growth in pregnant women with early FGR.

FGR and the potential therapeutic sanctions are of paramount importance for maternal-fetal medicine.

The objectives of this research, the methods, and the results are clearly stated.

But to get its full audience, it needs to be improved.

  1. Abstract, lines 15,16: “Results: fetal weight was higher at the beginning, after 1 and 2 weeks of observation in optimal vit. D group compared with suboptimal group.” The results should be presented in more detail, and the fetal weight should be discussed separately, as the fetal weight differed significantly since the enrolment, mainly due to the gestational age.
  2. Introduction, 24-25: “The low birth weight is responsible for 69.6% of neonatal deaths and 66.4% of intrauterine deaths. Fetal growth restriction is associated with intrauterine deaths, neonatal deaths (it is responsible for ca. 10% cases of perinatal deaths [1]”. This does not make sense. Also, low birth weight is not a death cause, but the condition that is associated with low birth weight.
  3. Introduction, lines 76-79: “A relationship between vitamin D supplementation in pregnancy complicated by FGR and the growth still remains controversial. Although numerous reports suggest a positive effect of supplementation with this vitamin, some of them do not confirm this relationship or give unambiguous results.”. Please, insert citations.
  4. Introduction, lines 76-79: “give unambiguous results”. The authors meant ambiguous?
  5. Results, lines 239-240, 243-244: “, for assumed values of factors”. Is this matching the groups? Because the groups should be matched for the characteristics that may influence fetal weight.
  6. Discussion, lines 377-380: CPR, UA, MCA, AEDV, REDV… the acronyms have to be explained in the text
  7. Discussion, line 380: “and this is shown in Table 4”. This is not what Table 4 is showing.
  8. Discussion, lines 393-394: “higher fetal weights were observed both after one and two weeks of observations for women supplementing vitamin D”. It should be omitted or the authors should comment that this is not an adequate indicator, because at the moment of inclusion into the study fetal weight differed significantly between the two groups; and one of the main causes was the pregnancy age.
  9. Why the authors did not search for Doppler (uterine, umbilical, cerebral) differences as well between the two groups? So, we can have an image and a justification for their findings?
  10. Was the biophysical score noted during the evaluations? Do we have proof regarding fetal wellbeing?
  11. What were the differences regarding gestational age at birth?
  12. Was the fetal weight on an ascending trend until birth?
  13. What was the outcome of the fetuses in the two groups?

In my view, these last questions are of great importance to establish the potential therapeutic effect of vitamin D correct supplementation and not a temporary increase of fetal weight.

Round 2

Reviewer 2 Report

In my view, the last questions from my first review are of great importance to establish the potential therapeutic effect of vitamin D correct supplementation. Especially regarding the potentially better outcome of the newborn and higher gestational age at birth.

In my view, the authors could still check the hospitals’ data for these variables which would greatly improve the communication.

Author Response

Dear Reviewer,

thank you for you work. We tried to come back to these data. Unfortunately key for removing the anonymity from id were lost. Unfortunately we cannot check patient outcome because we have only numbers.

This work was an initial study for further investigations. We didn't plan to check birth weight during planning the study.